# Exploratory Metabolomic and Lipidomic Profiling in a Manganese-Exposed Parkinsonism-Affected Population in Northern Italy

**DOI:** 10.3390/metabo15070487

**Published:** 2025-07-20

**Authors:** Freeman Lewis, Daniel Shoieb, Somaiyeh Azmoun, Elena Colicino, Yan Jin, Jinhua Chi, Hari Krishnamurthy, Donatella Placidi, Alessandro Padovani, Andrea Pilotto, Fulvio Pepe, Marinella Tula, Patrizia Crippa, Xuexia Wang, Haiwei Gu, Roberto Lucchini

**Affiliations:** 1Department of Environmental Health Sciences, Florida International University, Miami, FL 33199, USA; sazmoun@fiu.edu (S.A.); yan.jin@stjude.org (Y.J.); jinhua.chi@asu.edu (J.C.); haiweigu@asu.edu (H.G.); rlucchin@fiu.edu (R.L.); 2Department of Medical and Surgical Specialties, Radiological Sciences and Public Health, University of Brescia, 25123 Brescia, Italy; d.shoieb@studenti.unibs.it (D.S.); donatella.placidi@unibs.it (D.P.); 3Department of Environmental Medicine and Climate Science, Icahn School of Medicine at Mount Sinai, New York, NY 10029, USA; elena.colicino@mssm.edu; 4Department of Biomedical Engineering, Vibrant Sciences LLC., San Carlos, CA 95054, USA; hari@vibrantsci.com; 5Department of Clinical and Experimental Sciences, University of Brescia, 25123 Brescia, Italy; alessandro.padovani@unibs.it (A.P.); pilottoandreae@gmail.com (A.P.); 6Clinic of Neurology, Poliambulanza Foundation, 25124 Brescia, Italy; fulvio.pepe@poliambulanza.it; 7Clinic of Neurology, Esine Hospital of Valcamonica, 25040 Esine, Italy; marinellaturla@gmail.com; 8Teresa Camplani Foundation, Domus Salutis, 25123 Brescia, Italy; patrizia.crippa@ancelle.it; 9Department of Biostatistics, Florida International University, Miami, FL 33199, USA; xuexwang@fiu.edu; 10Dipartimento di Scienze Biomediche, Metaboliche e Neuroscienze, Sede Ex-Sc. Biomediche, University Modena and Reggio Emilia, 41121 Modena, Italy

**Keywords:** manganese exposure, Parkinsonism, metabolomics, lipidomics, environmental neurotoxicology, biomonitoring

## Abstract

Background/Objectives: Chronic manganese (Mn) exposure is a recognized environmental contributor to Parkinsonian syndromes, including Mn-induced Parkinsonism (MnIP). This study aimed to evaluate whole-blood Mn levels and investigate disease/exposure-status-related alterations in metabolomic and lipidomic profiles. Methods: A case–control study (N = 97) was conducted in Brescia, Italy, stratifying participants by Parkinsonism diagnosis and residential Mn exposure. Whole-blood Mn was quantified using ICP-MS. Untargeted metabolomic and lipidomic profiling was conducted using LC-MS. Statistical analyses included Mann–Whitney U tests, conditional logistic regression, ANCOVA, and pathway analysis. Results: Whole-blood Mn levels were significantly elevated in Parkinsonism cases vs. controls (median: 1.55 µg/dL [IQR: 0.75] vs. 1.02 µg/dL [IQR: 0.37]; *p* = 0.001), with Mn associated with increased odds of Parkinsonism (OR = 2.42, 95% CI: 1.13–5.17; *p* = 0.022). The disease effect metabolites included 3-sulfoxy-L-tyrosine (β = 1.12), formiminoglutamic acid (β = 0.99), and glyoxylic acid (β = 0.83); all FDR *p* < 0.001. The exposure effect was associated with elevated glycocholic acid (β = 0.51; FDR *p* = 0.006) and disrupted butanoate (Impact = 0.03; *p* = 0.004) and glutamate metabolism (*p* = 0.03). Additionally, SLC-mediated transmembrane transport was enriched (*p* = 0.003). The interaction effect identified palmitelaidic acid (β = 0.30; FDR *p* < 0.001), vitamin B6 metabolism (Impact = 0.08; *p* = 0.03), and glucose homeostasis pathways. In lipidomics, triacylglycerols and phosphatidylethanolamines were associated with the disease effect (e.g., TG(16:0_10:0_18:1), β = 0.79; FDR *p* < 0.01). Ferroptosis and endocannabinoid signaling were enriched in both disease and interaction effects, while sphingolipid metabolism was specific to the interaction effect. Conclusions: Mn exposure and Parkinsonism are associated with distinct metabolic and lipidomic perturbations. These findings support the utility of omics in identifying environmentally linked Parkinsonism biomarkers and mechanisms.

## 1. Introduction

In the modern era, characterized by unprecedented industrialization and urban development, the concept of the exposome has emerged as a pivotal framework for understanding the cumulative impact of occupational and environmental exposures on human health [1]. By emphasizing lifelong, cumulative exposures, the exposome framework recognizes that complex, interacting factors can negatively impact health outcomes later in life [1]. This perspective has catalyzed a re-evaluation of environmental toxicant mixtures and their additive roles in neurotoxicity development. Among these toxicants, manganese (Mn) has garnered increasing attention due to its widespread industrial use and growing recognition as a neurotoxicant [2].

In a 2021 survey conducted under the U.S. Environmental Protection Agency’s (EPA) Fourth Unregulated Contaminant Monitoring Rule (UCMR 4), Mnwas identified as an emerging public health concern [3]. Between 2018 and 2020, the EPA collected 37,963 water samples from 5034 public water systems across the United States. Mn was detected in 4527 of these systems (90%) at or above the EPA’s Minimum Reporting Level (MRL) of 0.4 µg/L, with concentrations ranging from 0.4 to 3960 µg/L [3]. Notably, 106 systems (2.1%) reported Mn concentrations exceeding the EPA’s non-enforceable Health Advisory Level of 300 µg/L, a threshold above which chronic exposure may lead to cognitive and motor impairments [3]. In contrast, other agencies have adopted more protective limits to address potential health risks. For example, Health Canada has set a Maximum Acceptable Concentration (MAC) of 120 µg/L, specifically to protect infants and young children from neurodevelopmental delay [4]. Nevertheless, a national survey of First Nations communities in Canada found that 4.0% of households exceeded Health Canada’s MAC [5]. These findings illustrate that elevated Mn exposure is occurring in real-world settings and may pose health risks to susceptible populations. Furthermore, as Mn becomes an increasingly sought-after heavy metal to meet the growing demand for electric vehicle (EV) batteries, the potential for environmental contamination and human exposure are likely to rise [6,7,8]. Advancing our understanding of the neurotoxic effects of Mn is essential to mitigating the potential health risks associated with its exposure.

Clinically, the neurotoxic potential of Mn is well documented, with evidence consistently linking Mn exposure to adverse cognitive and motor outcomes [9,10,11]. Experimental studies and epidemiological data show that chronically exposed populations are at elevated risk for Parkinsonian syndromes, a broader clinical category that includes Parkinson’s disease and atypical forms such as Mn-induced Parkinsonism (MnIP) [12,13,14]. Parkinson’s disease is a progressive neurodegenerative disorder that is primarily characterized by dopaminergic neuron loss in the substantia nigra and typically responds well to levodopa therapy. In contrast, MnIP results from manganese accumulation in basal ganglia structures, such as the globus pallidus, and often presents with gait disturbances, bradykinesia, and postural instability. However, it is typically non-responsive to dopaminergic treatment [15]. This clinical distinction is critical in understanding Mn as an exogenous trigger of Parkinsonism (PD) [15].

In response to the growing body of evidence of Mn neurotoxicity, researchers have investigated biomarkers of Mn exposure and its neurotoxic effects. For example, neuroimaging studies have examined reduced caudate FDOPA uptake via [18F] fluoro-L-DOPA PET, changes in GABA levels using magnetic resonance spectroscopy (MRS), and altered intrinsic functional connectivity on resting-state fMRI in Mn-exposed individuals. Biofluid and tissue-based biomarkers such as the Mn levels in blood, hair, nails, and teeth have also been examined, each reflecting different exposure windows and degrees of neurological impact [16]. Although biofluid analyses and neuroimaging techniques have been tested; reliable, reproducible, and accessible biomarkers remain elusive [17]. As a complementary strategy, metabolomics, a high-throughput technique for profiling small-molecule metabolites, has emerged as a powerful tool [18]. When combined with machine learning, metabolomics holds unique potential to identify novel biomarkers of Mn exposure and elucidate the biological pathways through which Mn exerts its toxicity [19,20,21]. This system-level approach may overcome the limitations of conventional biomarkers, which often fail to capture chronic exposure or early-stage dysfunction [22].

Building on the promise of metabolomics to uncover biomarkers of Mn toxicity, Brescia, Italy, offers a unique setting to explore its utility in the context of PD. A city with a history of heavy industrial activity, particularly in the steel and ferroalloy sectors, Brescia provides a natural context for studying environmental Mn exposure and its health effects. Previous investigations in this region found a higher prevalence of PD among municipalities closer to ferromanganese plants (492/100,000) compared to other province municipalities (321/100,000) [23]. Additionally, a significant association has been found between Bayesian Standardized Morbidity Ratios (SMRs) for PD and the concentrations of Mn dust [23]. These findings, along with the identification of genetic risk factors linking disruption in Mn intracellular homeostasis to PD, lay the foundation for our study [23].

This body of work aims to elucidate the metabolomic and lipidomic alterations associated with chronic Mn exposure and Parkinsonian outcomes. Using ultra-high-performance liquid chromatography–mass spectrometry (UHPLC-MS), we seek to identify the novel biomarkers and biological pathways influenced by Mn toxicity. Our hypothesis is that lifetime Mn exposure, particularly in industrial regions , is associated with distinct shifts in metabolomic and lipidomic profiles. By integrating exposomic principles and system-level analytics, this investigation seeks to advance our understanding of Mn-induced neurotoxicity and inform future therapeutic and public health interventions.

## 2. Materials and Methods

### 2.1. Study Population

Study participant selection methodology has been described previously [23]. Briefly, this study population includes 97 subjects divided into two groups: 48 exposed subjects (23 cases and 25 controls) and 49 non-exposed subjects (26 cases and 23 controls). This study is a subset of a larger cohort of approximately 1000 participants. The exposure level was determined based on the proximity to ferromanganese plants using Geographical Information Systems (GIS). This assessment included detailed spatial analysis to measure the distance of subjects’ residences from the ferromanganese plants. Exposed subjects lived in Val Camonica, at a distance ranging from 0 to 10 km from the three ferroalloy plants, whereas the non-exposed individuals lived in other areas of the Brescia province at distances ranging from 50 to 100 km from the Val Camonica ferroalloy plants. Additionally, environmental measures were considered, including the concentrations of Mn in soil samples, levels of airborne particles, and deposited dust in the areas surrounding the plants [24]. Length of residence was a key selection criterion for both exposed and non-exposed groups. Exposed subjects were those who had resided exclusively in Val Camonica or Bagnolo Mella, Italy, for their entire lives up until the date of sample collection, representing continuous lifetime exposure to industrial metals. In contrast, non-exposed subjects were selected from Lake Garda and Brescia city, Italy, non-industrial reference areas within the same province, but having minimal environmental heavy metal exposure [24]. The case inclusion criteria included individuals with a clinical diagnosis of Parkinsonism, defined as the presence of at least two of the following core motor features: bradykinesia, akinesia, rigidity, tremor, or postural instability [25]. This encompassed a broad classification of Parkinsonism, not limited to idiopathic Parkinson’s disease. We included patients with diagnoses of dementia with Lewy bodies [26], progressive supranuclear palsy [27], corticobasal syndrome [28], and multiple system atrophy [29]. Exclusion criteria included any history of iatrogenic or drug-induced Parkinsonism, traumatic Parkinsonism, or secondary Parkinsonism due to structural brain lesions such as tumors or encephalitis. Additionally, individuals with occupational exposure to Mn or other heavy metals (e.g., local ferroalloy workers, welders, battery plant workers) were excluded to minimize confounding from occupational sources of exposure. Control participants were randomly selected from non-neurological departments within the same hospitals, including dermatology, internal medicine, neurosurgery, ophthalmology, and orthopedics, to ensure comparable geographic representation and minimize selection bias. We also screened for and excluded controls with any known neurodegenerative conditions or parkinsonian symptoms. All selected subjects underwent fasting whole-blood sampling (0.2 mL/sample) for blood biomarker analyses at final enrollment. Self-reported data through a questionnaire covering demographics (age and sex), lifestyle habits (coffee consumption, alcohol consumption, smoking status, and prior diagnosis of comorbid diseases), and, exclusively to the cases, clinical diagnosis, treatment, and age at onset was collected. A visual representation of the study design can be found in Figure 1.

### 2.2. Sample Preparation

Whole-blood samples were collected in evacuated blood collection tubes containing spray-coated K_2_EDTA, the anticoagulant of choice for hematological determinations and whole-blood biomarker analyses. Following collection, tubes were inverted 8–10 times to ensure thorough mixing of blood and anticoagulant. Samples were stored at –80 °C by the University of Brescia (UniBS) until analysis. Upon obtaining participants’ consent and following the co-signing of a Material Transfer Agreement and a Data Use Agreement compliant with the European General Data Protection Regulation by both institutions and the approval of a new IRB (IRB-22–0246-AM03), biological samples and datasets were shared with the Department of Environmental Health Sciences at the Florida International University Robert Stempel College of Public Health and Social Work in 2023. The sample preparation and untargeted metabolomics and lipidomics methodology was described previously [30,31,32,33,34]. Briefly, whole-blood samples were thawed at 4 °C, vortexed for 5 s, and 50 µL of blood was transferred to a 1.5 mL Eppendorf tube. To this tube, 550 µL of methanol (MeOH) containing 200 µM 13C3-Lactate and 50 µM 13C5–15N-Glutamic Acid were added as the internal standards (ISs). The mixture was vortexed for 5 s and stored at −20 °C for 20 min. It was then centrifuged at 14,000 rpm for 10 min, and 450 µL of the supernatant was collected. This sample was dried using an Eppendorf Vacufuge drier (Eppendorf, Hamburg, Germany) for 120 min, reconstituted in 150 µL of H2O/PBS/ACN (2:2:6) solution, and centrifuged again at 14,000 rpm for 10 min. A volume of 100 µL of the supernatant was transferred to a glass vial for metabolomics analysis. The remaining supernatant from each sample was allocated to a new tube to serve as a pooled quality control (QC) sample. For lipidomics, starting with 50 µL of blood, 50 µL of 10X diluted PBS and 40 µL of 20X diluted Splash (Internal Standard Mixture, Avanti Polar Lipids, Alabaster, AL, USA) in MeOH were added. Then, 200 µL of Methyl tert-butyl ether (MTBE) was introduced to each sample in a ratio of MTBE/MeOH/H2O (10:2:5, *v*/*v*/*v*). The mixture was vortexed for 30 s and incubated at −20 °C for 20 min, followed by centrifugation at 14,000 rpm for 10 min to induce phase separation. The upper MTBE layer (150 µL) was carefully pipetted into a new Eppendorf tube. The tubes were left open in a hood for 2 h to dry. Afterward, 200 µL of isopropanol/MeOH (1:1) was added to the dried residues and the mixture briefly sonicated and centrifuged at 14,000 rpm for 10 min. A volume of 150 µL from each sample was then gently transferred to a glass vial for lipidomics analysis.

### 2.3. Manganese (Mn) Concentration Data Acquisition and Data Analysis

Whole-blood manganese concentrations were quantified using inductively coupled plasma mass spectrometry (ICP-MS) following a validated laboratory-developed protocol performed at Vibrant America Clinical Laboratories (Santa Clara, CA, USA). Whole-blood samples were collected using trace-element-free tubes and stored at 4 °C until analysis. For each measurement, 400 µL of whole blood was added to 3.6 mL of ICP-MS diluent consisting of 1 L of 18 MΩ deionized water, 20 mL of Optima-grade nitric acid (Fisher Scientific, Cat. No. A4671, Waltham, MA, USA), 20 mL of methanol, 0.5 mL of Triton X-100 (Sigma, Cat. No. 79284, St. Louis, MI, USA), and 1 mL of internal standard solution (Environmental Standard Mix 6, PerkinElmer, Cat. No. N9307808, Waltham, MA, USA). The mixture was vortexed to ensure complete lysis and homogenization. Calibration was performed using a single-element manganese standard (PerkinElmer TruQ™, Cat. No. N9303783, Waltham, MA, USA) at a certified concentration of 1000 µg/mL in 2% HNO_3_. Serial dilutions of the standard were prepared to generate a six-point calibration curve ranging from low ppb to high-ppb levels and verified using quality control samples (Bio-Rad Lyphochek Urine Metals Control Levels 1 and 2; Cat. Nos. 400 and 405, Hercules, CA, USA). Samples and calibrators were introduced into a PerkinElmer NexION ICP-MS equipped with an ESI SC-4 DX autosampler and operated using Syngistix software4.0. The instrument was tuned daily using a certified tuning solution (PerkinElmer, Cat. No. N8145051, Waltham, MA, USA), and optimization included checks for mass calibration, sensitivity, and oxide ratios using the SmartTune Manual workflow 4.0. The plasma was ignited and stabilized for 30 min prior to analysis, with argon gas pressure maintained at 100 psi. Sample introduction was performed via peristaltic pump using acid-resistant tubing, and the system was flushed between samples using 18 MΩ water rinses. Manganese detection was carried out at its most abundant isotope (^55Mn), and quantification was performed via internal standard correction and external calibration. All samples, standards, and quality controls were run in duplicate, and only runs meeting acceptance criteria for calibration linearity and control recovery were included in final data analysis. Data were exported using Syngistix and reviewed under the laboratory’s CLIA-certified workflow. Descriptive statistics—including mean, standard deviation (SD), median, interquartile range (IQR), minimum, and maximum—were calculated for each group. Differences in manganese concentrations between the Parkinsonism and control groups were assessed using the non-parametric Mann–Whitney U test. Conditional logistic regression was performed using the survival package to evaluate the association between whole-blood manganese concentration (µg/dL) and Parkinsonism status. The conditional logistic regression model was based on 32 matched case–control pairs (*n*= 64), matched on age and sex, and estimated β coefficients, odds ratios (ORs), 95% confidence intervals (CIs), and *p*-values. Statistical significance was defined as a two-sided *p*-value < 0.05. All statistical analyses were conducted using RStudio (Version 2024.12.1+563) with R (R Foundation for Statistical Computing, Vienna, Austria).

### 2.4. Untargeted Metabolomic and Lipidomic Data Acquisition

Mass spectrometry experiments were conducted using a Thermo UPLC-Exploris 240 Orbitrap MS (Waltham, MA, USA) as described previously [30,31,32,33,34]. Each sample was analyzed through dual injections in both positive and negative ionization modes, with an injection volume of 1 μL. For metabolomics, chromatographic separation was performed using a Waters XBridge BEH Amide column (150 × 2.1 mm, 2.5 μm, Waters Corporation, Milford, MA, USA) for both ionization modes. The system operated at a flow rate of 0.3 mL/min, while the auto-sampler was maintained at 4 °C and the column oven was set to 40 °C. The mobile phase consisted of Solvent A (0.1% formic acid in 95% H_2_O/5% acetonitrile (ACN)) and Solvent B (0.1% formic acid in 95% ACN/5% H_2_O). The elution profile included an initial 0.5 min hold at 90% B, followed by a decrease to 40% B at 10.5 min, which was held for 2 min and then returned to 90% B to accommodate subsequent injections. For lipidomics, chromatographic separation was conducted using a Waters XSelect HSS T3 column (Waters Corporation, Milford, MA, USA) in both ionization modes, at a flow rate of 0.3 mL/min. The mobile phase consisted of Solvent A (0.1% formic acid in 60% H_2_O/40% ACN) and Solvent B (0.1% formic acid in 90% isopropanol/10% ACN). The gradient began with an isocratic elution of 50% B for 3 min, which was gradually increased to 100% B over 12 min, held for 10 min, and then decreased to 50% B before the next sample injection. For mass spectrometric analysis, an electrospray ionization (ESI) source was used, with metabolomics data collected in the 70–1000 m/z range and lipidomics data collected in the 200–2000 m/z range.

### 2.5. Untargeted Metabolomic and Lipidomic Data Processing and Feature Retention Criteria

Untargeted metabolomics data were processed using a Thermo Compound Discoverer 3.3, with peak identification performed using HMDB, METLIN, mzCloud, Metabolika, ChemSpider, and an in-house database of approximately 300 metabolites listed in Appendix A. A list of the peak detection was conducted using a mass tolerance of 10 ppm, a peak intensity threshold of 100,000, and a chromatographic signal-to-noise (S/N) threshold of 1.5. Peak alignment was performed with an RT tolerance of 0.2 min, and feature grouping was conducted using a mass tolerance of 10 ppm. Untargeted lipidomics data were processed using Thermo LipidSearch 4.2, with precursor ion tolerance set at 5 ppm and product ion tolerance at 10 ppm. The m-score threshold was set at 2.0 for lipid identification and 5.0 for lipid alignment, while a retention time tolerance of 0.2 min was applied for lipid alignment. Lipid identifications were validated using MS/MS fragmentation patterns matched against the LipidMaps database. Peaks were considered true and retained for statistical analysis only if they were present in at least 80% of all samples and had a coefficient of variation (CV) below 20% across QC samples. Any peaks that failed to meet these criteria were excluded from further statistical analysis. Peak detection rates and feature retention were determined based on confidence levels as proposed by Schymanski et al. (2014) [35]. Using this framework, detected features were assigned to one of four levels:•Level 1: Confirmed structures with MS, MS/MS, and retention time matching reference standards.•Level 2: Probable structures, where spectral evidence suggests an exact molecular identity but without confirmation by reference standards.•Level 3: Tentative candidates, where multiple potential structures exist, but insufficient evidence prevents assigning a single structure.•Level 4: Unequivocal molecular formulas, where the exact mass and isotopic patterns confirm the chemical formula, but no specific structure is proposed.

### 2.6. Statistical Power and Sample Size Justification

Given the exploratory nature of this study and its high-dimensional design, we conducted a power analysis to assess whether our sample size was broadly sufficient to detect meaningful group-level differences in metabolomic and lipidomic profiles. The preliminary analysis, based on variance estimates from a subset of pilot metabolomic data collected under similar conditions, suggested that 16 subjects per group would provide 90% power to detect differences at a 5% false discovery rate (FDR). This calculation assumed a conservative effect size (meandifferencetostandard deviation ratio) of 0.542, which aligns with the median values observed in prior untargeted omics studies. We recognize that a one-way omnibus test is a simplification and may not fully capture the structure of a stratified 2 × 2 factorial design. However, this approximation was used to estimate a baseline threshold for power feasibility, not to test formal hypotheses. With 97 participants evenly distributed across four groups, the study design remains within the typical range used in comparable exploratory metabolomic and environmental neuroepidemiology studies. All power calculations were conducted in RStudio (Version 2023.09.1+494) using standard statistical methods for omics data.

### 2.7. Sociodemographic Data Analysis

We report descriptive statistics for all covariates of the models, which were selected for their biological role in the exposure or the disease. The covariates include age (continuous), sex, coffee consumption (yes/no), alcohol consumption (yes/no), smoking status (ever/never), and clinically diagnosed individual comorbidities (yes/no). Comorbidities included the following conditions: diabetes, stroke, hypertension, leukemia, heart disease, liver disease, kidney disease, and thyroid disease. We also collected and analyzed data on medication usage, including both Parkinson’s disease-specific therapies (e.g., levodopa, dopamine agonists, MAO-B inhibitors, COMT inhibitors) and non-Parkinson's disease medications (e.g., statins, beta blockers, calcium channel blockers, aspirin). Medication usage was recorded as binary variables (Yes/No), and Parkinson's disease-specific medications were only reported in the Parkinsonism group and were not applicable to control participants. We performed chi-squared tests for categorical variables and t-tests for continuous variables to compare groups regarding disease status and exposure status. All results were considered statistically significant with a Type I error of 5%.

### 2.8. Metabolomic Data Analysis

We used a 2 × 2 study design to investigate the metabolomic biomarker associations with (1) exposures (yes/no), (2) diagnosis of PD (yes/no), and (3) their interactions. We applied a series of computational techniques to analyze and interpret the data derived from 550 annotated metabolites. To address the skewness of metabolite concentration data and to meet the model assumptions, a log10-transformation was implemented. Next, we performed outlier identification utilizing unsupervised Principal Component Analysis (PCA) and supervised Partial Least Squares-Discriminant Analysis (PLS-DA). To statistically evaluate the PCA results, a Permutational Multivariate Analysis of Variance (PERMANOVA) was conducted. To assess the robustness of the PLS-DA model, 5-fold cross-validation metrics were calculated, permutation testing was performed, and Variable Importance in Projection (VIP) scores for each metabolite were analyzed. Metabolites with higher VIP scores were considered more influential in differentiating between subject groups. We further employed two-way Analysis of Covariance (ANCOVA) to discern the metabolomic alterations of (1) exposures (yes/no), (2) diagnosis of PD (yes/no), and (3) their interactions while adjusting for covariates (age and sex). Raw and adjusted *p*-values using the false discovery rate (FDR) Benjamini–Hochberg method were reported. Beta coefficients, 95% confidence intervals (CIs), and partial eta squared (n^2^_p_) were also reported. The (n^2^_p_) value represents the proportion of total variance in metabolite levels that is attributable to each main effect or their interaction, after accounting for other variables included in the model. (n^2^_p_) values were hierarchically ranked to ascertain a rank-ordered list of metabolites. Statistical significance was determined using a 5% FDR threshold. To evaluate the biological relevance of our significant features, we performed overrepresentation pathway analysis (OPA) and metabolite set enrichment analyses (MSEA) using statistically significant metabolites as our test set. Organism-specific pathway sets were used as our background set for pathway analysis (Homo Sapien-KEGG) and enrichment analysis (Homo Sapien-RaMP-DB), as suggested by previous reports [22,36]. Using a hypergeometric test for enrichment and relative-betweenness centrality for topology analysis, pathways were identified based on the statistical significance and impact score (Impact) associated with each main effect or the interaction between these factors. Pathways with an uncorrected Type I error of 5% were considered biologically relevant. The enrichment analysis utilized the GlobalTest and topology analysis via the relative-betweenness-centrality method, which identified metabolites that were significantly enriched, considering their position within the metabolic network. The ‘Expect’ column represents the expected number of metabolite hits in each set based on a random distribution, calculated under the null hypothesis. This value provides a baseline for comparing the observed number of hits, allowing for the assessment of whether the metabolite set is significantly enriched beyond what would be expected by chance. All statistical analyses and visualizations were performed using a combination of software tools including RStudio (Version 2023.09.1+494), MetaboAnalyst (Version 6), and Jupyter Notebooks (Version 6.5.4), accessed through Anaconda.

### 2.9. Lipidomic Data Analysis

We applied a series of computational techniques to analyze and interpret the data derived from 94 identified lipid species following the analysis methodology described above for metabolomic data. Note, biological relevance was assessed using an alternative software tool, LIPEA1.5.0. Briefly, LIPEA leverages the KEGG Database to perform OPA. OPA was conducted using Fisher’s exact test to identify statistically significant lipids. These lipids showed overrepresentation when compared to the KEGG Database list of lipids, which served as the background dataset [22,36]. The ‘Pathway Lipids’ column represents the number of lipids associated with each specific pathway that were tested for enrichment. To determine whether these pathways are significantly enriched with the lipids identified in our dataset, we applied Fisher’s exact test. This test assesses the association between the identified lipids and pathways, comparing the observed number of lipids within each pathway to what would be expected under a null hypothesis of no association. This background set included all lipids that could potentially be identified in the specific organism under study, Homo sapiens. Biologically relevant overrepresented pathways were identified using the 5% FDR threshold. Note, due to a lack of statistically significant features identified for the exposure effect, biological relevance was not assessed using OPA. All statistical analyses and visualizations were performed using a combination of software tools including RStudio (Version 2023.09.1+494), LIPEA (https://hyperlipea.org/home), accessed on 2 June 2024, and Jupyter Notebooks (Version 6.5.4), accessed through Anaconda.

## 3. Results

### 3.1. Demographic and Lifestyle Characteristics of the Study Population

Table 1 compares the characteristics of the study population across the two main groups—PD and controls—with each being further divided into “Exposed” and “Non-Exposed” categories. The control group included 48 participants (52.1% exposed, 47.9% non-exposed), and the PD group had 49 participants (46.9% exposed, 53.1% non-exposed). Mean ages ranged from 66.4 to 72.4 years, with no significant differences between groups. Males constituted 56.5% to 80.7% of participants, with no significant sex distribution differences. Coffee consumption, alcohol consumption, smoking status, individual comorbidities, and medication usage also showed no significant differences, suggesting no association with PD in this population. A detailed analysis of medication usage across groups is provided in Appendix A, where PD-specific and non-PD medications are compared. Geographically, non-exposed participants were from Brescia City and Lake Garda, while exposed participants were predominantly from Val Camonica, with a smaller number from Bagnolo Mella, underscoring a clear distinction between the non-exposed and exposed groups. It should be noted that a disproportionate number of participants lived in Val Camonica across both the control and PD groups.

### 3.2. Manganese Measurement Using Inductively Coupled Plasma Mass Spectrometry (ICP-MS)

Whole-blood manganese concentrations were significantly higher in individuals with Parkinsonism compared to controls (Figure 2). Among those with available blood manganese measurements, the median manganese level in the Parkinsonism group was 1.55 µg/dL (IQR = 0.75) versus 1.02 µg/dL (IQR = 0.37) in controls (Table 2). A Mann–Whitney U test revealed this difference was statistically significant (*p* = 0.0010). To further evaluate the association between manganese levels and Parkinsonism while accounting for potential confounding by age and sex, a conditional logistic regression analysis was conducted among 32 matched case–control pairs (*n* = 64 total; matching on age and sex). Higher manganese levels were associated with increased odds of Parkinsonism (β = 0.88), corresponding to an odds ratio (OR) of 2.42 (95% CI: 1.13–5.17; *p* = 0.0223) (Table 3). These findings suggest that elevated whole–blood manganese concentrations are significantly associated with increased odds of Parkinsonism in this study population. Additional analysis assessing the association between manganese levels and Parkinsonism can be found in Appendix A.

### 3.3. Machine-Learning-Based Metabolomic Exploratory Data Analysis

Principal Component Analysis (PCA) and Partial Least Squares Discriminant Analysis (PLS-DA) were performed to identify patterns in the metabolomic data and distinguish between control and Parkinsonism (PD) subjects, further stratified by exposure status (exposed vs. non-exposed). The PCA scores plot (Appendix A) shows the first two principal components, with PC1 explaining 13.6% and PC2 explaining 9.7% of the total variance. A PERMANOVA analysis confirmed significant separation between groups (F = 22.665, R^2^ = 0.42234, *p* = 0.001, based on 999 permutations). The PLS-DA scores plot (Appendix A) revealed enhanced group separation compared to PCA, with Component 1 explaining 12.9% and Component 2 explaining 5% of the variance. As a supervised technique, PLS-DA used group labels to maximize separation, particularly distinguishing PD exposed from control non-exposed subjects, suggesting an influence of exposure status on the metabolomic profile associated with PD. Model performance was evaluated using five-fold cross-validation across models with one to five components (Appendix A). Metrics assessed included accuracy, R^2^, and Q^2^ values. Permutation testing with 1000 permutations using prediction accuracy as the test statistic resulted in a *p*-value < 0.001, indicating that the model’s performance was significantly better than expected by random chance. Additionally, assessment of separation distance (between-class variation, B/W) further confirmed statistically significant group separation (*p* < 0.001). The Variable Importance in Projection (VIP) scores ranked the top 20 metabolites contributing to group discrimination (Appendix A). Sulbactam exhibited the highest VIP score, followed by dopamine 3-O-sulfate and 4-amino-1-piperidinecarboxylic acid. Relative abundance heatmaps (Appendix A) demonstrated distinct metabolite patterns across groups, supporting the observed discrimination in multivariate analyses.

### 3.4. Analysis of Covariance (ANCOVA) and Metabolite Associations with Disease Effect

Following the exploratory analysis, we annotated 98 statistically significant metabolites related to disease status after FDR correction. The volcano plot, Figure 3A, contrasts the significance of observed changes in relative abundance (indicated by *p* < 0.05) with their effect size (represented by beta coefficients) for the disease effect. Notably, metabolites such as “sulbactam” (β = 1.61, 95% CI [1.32, 1.90], FDR *p* < 0.001), “dopamine 3-O-sulfate” (β = 1.15, 95% CI [0.92, 1.39], FDR *p* < 0.001), “3-(sulfooxy)-L-tyrosine” (β = 1.12, 95% CI [0.87, 1.37], FDR *p* < 0.001), and “vanillic acid 4-sulfate” (β = 0.87, 95% CI [0.62, 1.11], FDR *p* < 0.001) exhibited high statistical significance, suggesting a strong association with the disease effect. The full dataset is listed in Appendix A. The bar chart, Figure 3B, represents the top 25 metabolites associated with the disease effect, hierarchically ranked by their partial eta squared (η^2^_p_) values. Notably, “sulbactam” (η^2^_p_= 0.70, FDR *p* < 0.0001), “dopamine 3-O-sulfate” (η^2^_p_= 0.61, FDR *p* < 0.0001), “3-(sulfooxy)-L-tyrosine” (η^2^_p_ = 0.55, FDR *p* < 0.0001), and “vanillic acid 4-sulfate” (η^2^_p_ = 0.55, FDR *p* < 0.0001) demonstrated the largest magnitude of effect and strongest statistical significance. These findings suggest potential metabolomic signatures associated with disease status. The full dataset is provided in Appendix A. The pathway analysis bubble plot, Appendix A, evaluates the statistical significance and topological impact of metabolic pathways associated with the disease effect. Notably, “Alanine, aspartate and glutamate metabolism” (Impact = 0.05, *p* = 0.001), “Citrate cycle (TCA cycle)” (Impact = 0.15, *p* = 0.01), “Glyoxylate and dicarboxylate metabolism” (Impact = 0.31, *p* = 0.03), and “Arginine and proline metabolism” (Impact = 0.02, *p* = 0.03) exhibited nominal statistical significance. Additional pathway statistics are presented in Appendix A. Appendix A presents the results of the Metabolite Set Enrichment Analysis (MSEA), performed using the RaMP-DB pathway library. Enriched metabolite sets included “Alanine, aspartate and glutamate metabolism” (Expected hits = 0.352, *p* = 0.006) and “Citric acid cycle and respiratory electron transport” (Expected hits = 0.255, *p* = 0.002). The full MSEA results are available in Appendix A. Note that biological relevance should be interpreted with caution, as no pathway or enriched set achieved statistical significance after FDR correction. However, these biological processes may contribute to the etiology and pathogenesis of PD, warranting further mechanistic investigation.

### 3.5. ANCOVA and Metabolite Associations with Exposure Effect

We annotated 28 statistically significant metabolites related to exposure status after FDR correction. The volcano plot highlights several metabolites significantly associated with the exposure main effect. Notable examples include “L-iditol” (β = 0.87, 95% CI [0.55, 1.19], FDR *p* < 0.001), “D-(–)-mannitol” (β = 0.64, 95% CI [0.41, 0.87], FDR *p* < 0.001), “L-histidinol phosphate” (β = 0.59, 95% CI [0.33, 0.85], FDR *p* = 0.002), “Citric acid” (β = 0.51, 95% CI [0.28, 0.75], FDR *p* = 0.006), and “lyngbic acid” (β = –0.24, 95% CI [–0.34, –0.13], FDR *p* < 0.001). The full statistical results are presented in Appendix A. The bar chart in Appendix A displays the top metabolites associated with the exposure main effect, hierarchically ranked by partial eta squared (η^2^_p_) values. Among these, “tetrahomomethionine” (η^2^_p_ = 0.221, FDR *p* < 0.01), “lafutidine” (η^2^_p_ = 0.220, FDR *p* < 0.001), “D-(–)-mannitol” (η^2^_p_ = 0.178, FDR *p* < 0.05), and “glycocholic acid” (η^2^_p_ = 0.172, FDR *p* > 0.05) demonstrated the largest effect sizes in relation to the exposure effect. The full ranked list of exposure-associated metabolites and associated statistics is provided in Appendix A. The pathway analysis bubble plot (Figure 4B) reveled “alanine, aspartate and glutamate metabolism” (Impact = 0.2, *p* < 0.001), “butanoate metabolism” (Impact = 0.03, *p* = 0.004), and “glyoxylate and dicarboxylate metabolism” (Impact = 0.03, *p* = 0.01) as statistically significant, indicating the largest effect of exposure on these pathways. Additionally, Appendix A represents the top 25 metabolite sets enriched due to exposure, as determined by MSEA. The results of the enrichment analysis revealed several metabolites sets with significant alterations related to transmembrane transporters. “SLC-mediated transmembrane transport” (Expected hits = 0.326, *p* = 0.003), “transport of small molecules” (Expected hits = 0.437, *p* = 0.007), and “sodium-coupled sulphate, di- and tri-carboxylate transporters” (Expect = 0.00841, *p* = 0.008) showed higher levels of enrichment, indicating perturbations in these metabolite sets. Note that biological relevance should be interpreted with caution, as no pathway or enriched set achieved statistical significance after FDR correction. However, these biological processes may reflect biological pathways responsive to Mn exposure.

### 3.6. ANCOVA and Metabolite Associations with Interaction Effect

We annotated 59 statistically significant metabolites associated with the interaction effect between disease status and exposure status after FDR correction. The volcano plot (Figure 5A) highlights several metabolites significantly associated with the interaction effect between exposure and Parkinsonism status. Notable examples include “palmitelaidic acid” (β = 0.30, 95% CI [0.19, 0.41], FDR *p* < 0.001), “pentobarbital” (β = 0.36, 95% CI [0.21, 0.51], FDR *p* < 0.001), “(S)-2-methylbutanal” (β = 0.59, 95% CI [0.36, 0.83], FDR *p* < 0.001), “n-butyl lactate” (β = 0.59, 95% CI [0.36, 0.83], FDR *p* < 0.001), “lyngbic acid” (β = 0.36, 95% CI [0.21, 0.51], FDR *p* < 0.001), and “PC” (β = 0.97, 95% CI [0.56, 1.39], FDR *p* < 0.001). These metabolites exhibited both high statistical significance and substantial interaction effect sizes, implicating them in potential synergistic mechanisms between exposure and disease. The full statistical results are presented in Appendix A. The bar chart in Appendix A displays the top metabolites associated with the interaction effect, ranked by their partial eta squared (η^2^_p_) values. Among these, “palmitelaidic acid” (η^2^_p_ = 0.24, FDR *p* < 0.001), “pentobarbital” (η^2^_p_ = 0.21, FDR *p* < 0.001), “(S)-2-methylbutanal” (η^2^_p_ = 0.21, FDR *p* < 0.001), “lyngbic acid” (η^2^_p_ = 0.20, FDR *p* < 0.001), and “PC”, (η^2^_p_ = 0.19, FDR *p* < 0.001) demonstrated the strongest effect sizes. The full results are provided in Appendix A. The pathway analysis bubble plot in Appendix A revealed “vitamin B6 metabolism” (Impact = 0.08, *p* = 0.03) as significant. Additionally, MSEA (Figure 5B) generally identified amino acid metabolism and transmembrane transporters as enriched metabolite sets. These findings corroborate what was previously found in our main effect of disease and our main effect of exposure. The top three statistically significant enriched sets were “phase II—Conjugation of compounds” (Expected hits = 0.341, *p* < 0.001), “sudden infant death syndrome (SIDS), susceptibility pathways” (Expected hits = 0.0327, *p* < 0.001), and “SLC transporter disorders” (Expected hits = 0.187, *p* < 0.001). However, the smaller size of their bubbles indicates a low enrichment ratio. Conversely, high enrichment ratios were found among “defective SLC22A12 causes renal hypouricemia 1 (RHUC1)” (Expected hits = 0.00467, *p* = 0.005), “defective SLC6A3 causes Parkinsonism-dystonia infantile (PKDYS)” (Expected hits = 0.00467, *p* = 0.005), and “defective SLC35A1 causes congenital disorder of glycosylation 2F (CDG2F)” (Expected hits = 0.00467, *p* = 0.005). Note that biological relevance should be interpreted with caution as no pathway or enriched set achieved statistical significance after FDR correction. Additional data are listed in Appendix A.

### 3.7. ANCOVA and Lipid Associations with Disease Effect, Exposure Effect, and Interaction Effect

Transitioning from our metabolomic analysis, our ANCOVA annotated 33 lipid species that were significantly associated with the disease effect after FDR correction. The volcano plot in Figure 6A shows lipid species such as “TG(16:0_10:0_18:1)” (β = 0.79, 95% CI [0.45, 1.14], FDR *p* < 0.01), “SM(d44:3)” (β = 0.17, 95% CI [0.08, 0.27], FDR *p* < 0.05), “PE(18:0p_18:1)” (β = 0.16, 95% CI [0.07, 0.25], FDR *p* < 0.05), “TG(16:0_14:0_16:0)” (β = 0.33, 95% CI [0.15, 0.51], FDR *p* < 0.05), and “TG(18:0_18:0_18:1)” (β = 0.38, 95% CI [0.18, 0.57], FDR *p* < 0.05). The full results are available in Appendix A. The bar chart in Figure 6B ranks the top 25 lipids by (η^2^_p_) values, highlighting the largest effects associated with the disease effect. Notable lipid species include “TG(16:0_10:0_18:1)” (η^2^_p_ = 0.41, FDR *p* < 0.01), “TG(16:1_14:0_18:2)” (η^2^_p_ = 0.28, FDR *p* < 0.05), “TG(16:0_14:0_16:0)” (η^2^_p_ = 0.27, FDR *p* < 0.05), “TG(16:0_14:0_14:0)” (η^2^_p_ = 0.27, FDR *p* < 0.05), “TG(16:0_14:0_18:2)” (η^2^_p_ = 0.26, FDR *p* < 0.05), and “TG(18:0_18:0_18:1)” (η^2^_p_ = 0.23, FDR *p* < 0.05). Additional data are listed in Appendix A. For the exposure effect, no lipid species were statistically significant after FDR correction; however, many lipid species demonstrated negative beta coefficients, suggesting reduced lipid levels with exposure. Notable examples include “Cer(d18:1_24:0)” (β = −0.09, 95% CI [−0.15, −0.03], FDR *p* = 0.25), “Cer(d18:0_24:0)” (β = −0.12, 95% CI [−0.20, −0.04], FDR *p* = 0.25), “LPC(18:0)” (β = −0.13, 95% CI [−0.22, −0.03], FDR *p* = 0.30), “Cer(d18:1_24:1)” (β = −0.09, 95% CI [−0.16, −0.02], FDR *p* = 0.43), and “Hex1Cer(d18:1_24:0)” (β = −0.10, 95% CI [−0.20, −0.01], FDR *p* = 0.46). The full results are available in Appendix A. The interaction effect annotated 12 statistically significant lipid species after FDR correction (Figure 6C), all of which belonged to either the phosphatidylethanolamine (PE) or phosphatidylcholine (PC) lipid classes. Notable examples include “PE(16:0_20:4)” (β = −0.40, 95% CI [−0.62, −0.17], FDR *p* = 0.02), “PE(40:7e)” (β = −0.38, 95% CI [−0.61, −0.16], FDR *p* = 0.02), “PC(35:4)” (β = −0.26, 95% CI [−0.41, −0.10], FDR *p* = 0.02), “PE(18:0_20:4)” (β = −0.26, 95% CI [−0.41, −0.10], FDR *p* = 0.02), and “PE(40:5e)” (β = −0.46, 95% CI [−0.73, −0.20], FDR *p* = 0.02), “PC(37:5e)” (β = −0.48, 95% CI [−0.76, −0.21], FDR *p* = 0.02), “PE(38:5e)” (β = −0.34, 95% CI [−0.54, −0.13], FDR *p* = 0.02), “PE(18:0p_22:4)” (β = −0.44, 95% CI [−0.71, −0.17], FDR *p* = 0.02), “PC(37:6)” (β = −0.33, 95% CI [−0.53, −0.12], FDR *p* = 0.02), “PE(16:0p_20:4)” (β = −0.30, 95% CI [−0.49, −0.10], FDR *p* = 0.03), and “PE(36:5e)” (β = −0.26, 95% CI [−0.44, −0.09], FDR *p* = 0.03). The bar chart in Figure 6D ranks the top 25 lipid species by (η^2^_p_) values, with “PC(37:5e)” (η^2^_p_ = 0.12, FDR *p* = 0.02), “PE(40:5e)” (η^2^_p_ = 0.12, FDR *p* = 0.02), “PE(16:0_20:4)” (η^2^_p_ = 0.12, FDR *p* = 0.02), “PE(40:7e)” (η^2^_p_ = 0.11, FDR *p* = 0.02), “PC(35:4)” and “PE(18:0_20:4)” (η^2^_p_ = 0.11 each, FDR *p* = 0.02), “PE(18:0p_22:4)” (η^2^_p_ = 0.10, FDR *p* = 0.02), “PE(38:5e)” (η^2^_p_ = 0.10, FDR *p* = 0.02), “PC(37:6)” (η^2^_p_ = 0.10, FDR *p* = 0.02), “PE(16:0p_20:4)” (η^2^_p_ = 0.09, FDR *p* = 0.03), and “PE(36:5e)” (η^2^_p_ = 0.09, FDR *p* = 0.03) demonstrating the largest effect sizes. Again, all of these belonged to either the phosphatidylethanolamine (PE) or phosphatidylcholine (PC) lipid classes. The pathway analysis in Appendix A associated the disease effect to retrograde endocannabinoid signaling (8 matched lipids, Benjamini *p* = 0.028), and ferroptosis (11 matched lipids, Benjamini *p* = 0.028). The interaction effect was associated with retrograde endocannabinoid signaling (8 matched lipids, Benjamini *p* = 0.036), sphingolipid signaling pathway (9 matched lipids, Benjamini *p* = 0.036), and ferroptosis (11 matched lipids, Benjamini *p* = 0.037). Pathway analysis for the exposure effect was not included due to the absence of significant lipid species. Additional data are listed in Appendix A.

## 4. Discussion

These findings suggest that alterations in metabolite and lipid signatures may have the potential to serve as predictive biomarkers for specific PD subtypes related to environmental exposure. Additionally, these insights into the metabolomic and lipidomic perturbations associated with PD and environmental exposure could contribute to improved preventive screenings, novel diagnostic methods, and the development of future therapeutic strategies. While statistical significance is essential, biological relevance is equally important in identifying potential clinically relevant biomarkers. Therefore, we focus our discussion on metabolomic and lipidomic perturbations that are both statistically significant and biologically relevant, as represented in Table 4.

Metabolites related to medication usage, dietary habits, amino acid metabolism, cellular redox balance, and vitamin B regulation were identified as being significant in our study. However, the metabolites sulbactam, dopamine 3-O-sulfate (DA-3S), and vanillic acid 4-sulfate may be biologically artifactual. For example, sulbactam, a beta-lactamase inhibitor, is often used by PD patients to treat infections; furthermore, DA-3S, a dopamine metabolite, is an end product of levodopa metabolism in PD patients [37,38,39]. Similarly, vanillic acid 4-sulfate, found largely in virgin olive oil, is also not likely to serve as a diagnostic biomarker due to the dietary habits within our population [40]. Conversely, biologically relevant metabolites such as 3-sulfoxy-L-tyrosine, formiminoglutamic acid, glyoxylic acid, and 2-Hydroxy-3-[3-methoxy-4-(sulfooxy)phenyl] propanoic acid were identified. 3-Sulfoxy-L-tyrosine is formed via a post-translational modification of tyrosine and plays a role in norepinephrine synthesis, which may be reduced in PD [41]. Formiminoglutamic acid is an intermediate in L-histidine breakdown and serves as a biomarker for folate levels. Its increase can indicate vitamin B12 deficiency, which is associated with PD [42]. Glyoxylic acid is crucial for cellular redox balance and amino acid metabolism. This acid is linked to mitochondrial function and neuroprotection, highlighting its importance in PD [43]. Lastly, Aryl sulfates (like 2-Hydroxy-3-[3-methoxy-4-(sulfooxy)phenyl]propanoic acid) may be metabolic indicators of impaired sulfatide metabolism in PD patients, although there is limited information on this to date [44]. The disease effect also impacted pathways related to amino acid metabolism and the TCA cycle. Alterations in 3-sulfoxy-L-tyrosine, formiminoglutamic acid, and glyoxylic acid suggest that nerve degeneration and mitochondrial dysfunction could lead to changes in amino acid metabolism, exacerbating PD symptoms [45]. These findings align with the existing literature and underscore the complex interplay between PD and metabolic dysfunction.

The metabolites tetrahomomethionine, lafutidine, and mannitol were strongly associated with the exposure effect but were excluded as diagnostic or prognostic biomarkers due to their biological irrelevance. Tetrahomomethionine is linked to dietary intake of L-methionine, though its biosynthetic pathway is unclear [46]. Additionally, lafutidine, a histamine H2 receptor antagonist commonly used to reduce gastric acid secretion in the treatment of gastrointestinal ulcers, and mannitol, an osmotic diuretic used to manage fluid retention and cerebral edema, are related with medical treatment [47,48]. In contrast, glycocholic acid, a bile acid conjugated with glycine, was significantly upregulated, possibly due to liver damage caused by metal exposure, which aligns with its role in emulsifying fats and enhancing solubility [49]. Furthermore, disruptions in butanoate and glutamate metabolism may suggest impaired conversion of L-glutamate into GABA, a key inhibitory neurotransmitter, potentially contributing to the neurochemical imbalances observed in Mn exposure [50]. Specifically, Mn interferes with NMDA receptor (NMDAR) signaling by blocking calcium channels and downregulating GluN2B expression, impairing excitatory neurotransmission [51]. Concurrently, Mn reduces GABAA receptor expression while inducing GABAB receptor expression, further disrupting inhibitory signaling and promoting alpha-synuclein accumulation, a hallmark of PD. These disturbances may exacerbate glutamate–GABA imbalances, which are implicated in the neurodegenerative effects of Mn exposure [51]. Additionally, Mn exposure perturbed SLC-mediated transport, presumably by activating proinflammatory genes (e.g., IL-6, IL-1B, CCL2), competing with essential metal ions, and blocking TRPC3 channels in astrocytes [52]. Moreover, metals disrupt amino acid metabolism by binding to critical enzymatic sites, catalyzing oxidation, interfering with protein folding, and displacing essential ions [53]. These alterations may contribute to the observed perturbations in amino acid metabolism associated with our exposure effect.

The metabolites N-butyl lactate, 2-methylbutanal, and pentobarbital were statistically associated with the interaction effect between disease and exposure status but are likely artifactual due to diet and medication use. N-Butyl lactate and 2-methylbutanal are food additives, suggesting dietary exposure, while pentobarbital, a barbiturate used as a sedative and anticonvulsant, likely reflects prior pharmaceutical treatment rather than an endogenous metabolic process [54,55,56]. However, palmitelaidic acid was statistically significant and biologically relevant. Palmitelaidic acid, a trans fatty acid associated with increased cardiovascular disease risk, is primarily obtained through diet. It also plays a role in lipid peroxidation, a process where ROS causes the oxidative degradation of lipids, leading to oxidative stress, cell damage, and PD disease progression [57,58]. Key pathways affected by the interaction effect—glucose homeostasis, amino acid metabolism, and SLC transporter disorders—highlight how the combined exposure and disease effects mirror the biological perturbations seen in the independent main effects, with these disruptions presumably being heightened in the interaction effect. Uniquely, vitamin B6 metabolism was significantly impacted, highlighting its importance in amino acid metabolism and neurotransmitter synthesis, which are crucial for proper brain function [59]. Additionally, impairments in glucose homeostasis, supported by the previous PD literature, was associated with the interaction effect, potentially due to mechanisms like insulin resistance, oxidative stress, and blood–brain barrier dysfunction [60].

Lipid classes were significantly altered in association with the disease and interaction effects, while the exposure effect did not reach statistical significance. This may suggest a greater lipid class alteration when disease and exposure effects are combined. The key lipids involved include triglycerides (TGs), lysophosphatidylcholine (LPC), ceramides, and phosphatidylcholines (PCs), all of which are linked to cognitive function [61]. Longitudinal studies indicate that elevated TG levels increase the risk of cognitive impairment, a hallmark of late-stage PD [62]. Ceramides, a class of sphingolipids (SPs), are crucial for cellular processes like division, differentiation, and apoptosis. As such, dysregulation in sphingolipid metabolism is associated with neurological disorders and disease progression [63]. LPC, a phospholipid integral to cell membrane structure and function, plays roles in signaling, inflammation, and immune regulation. Abnormal LPC levels have been linked to neurological disorders and may reflect oxidative stress response. Interestingly, the interaction effect showed a downregulation in lipid classes, particularly phosphatidylethanolamine (PE), corroborating the findings from other PD studies [64,65,66,67,68,69]. Furthermore, these perturbed lipids are associated with pathways like retrograde endocannabinoid signaling, ferroptosis, and sphingolipid metabolism [64,65,66,67,68,69]. The disease and interaction effects identified both endocannabinoid signaling and ferroptosis as enriched pathways, with the interaction effect also identifying sphingolipid metabolism. Endocannabinoids, such as anandamide (AEA) and 2-arachidonoylglycerol (2AG), regulate synaptic activity and mitochondrial function through CB1 receptors [64,65,66,67,68,69]. Ferroptosis, a form of regulated cell death driven by ROS and lipid peroxidation, is involved in various pathological processes, including PD [64,65,66,67,68,69]. Lastly, sphingolipids, such as ceramide and sphingosine-1-phosphate (S1P), exert opposing effects on cell stress responses and survival, underscoring their critical roles in these pathways [64,65,66,67,68,69].

This study provides novel exploratory insights into the metabolomic and lipidomic alterations associated with manganese exposure and PD. We identified several potential biomarkers—3-sulfoxy-L-tyrosine, formiminoglutamic acid, glyoxylic acid, glycocholic acid, butanoate, glutamate, and palmitelaidic acid—that may aid in the early detection and management of MnIP. A key strength of this study is its case–control design, combined with advanced metabolomic and lipidomic analyses that enabled the identification of biomarkers linked to both exposure and disease effects. However, several limitations should be acknowledged. First, the cross-sectional design and prior PD diagnosis limit our ability to determine temporal relationships, and reverse causality cannot be ruled out. Additionally, although false discovery rate (FDR) corrections were applied at the feature level, many pathway interpretations did not remain significant after additional FDR correction. Therefore, the observed associations between exposure and metabolite profiles should be interpreted cautiously and as hypothesis-generating rather than causal. Expanding the scope of research to include transcriptomic and proteomic analyses will deepen our understanding of the molecular pathways disrupted by manganese exposure. Second, while prior research on this cohort identified alpha-synuclein as a genetic determinant, no new genetic analyses were performed [23]. Future studies will explore gene–environment interactions in the broader cohort to better characterize the interplay between genetic susceptibility and environmental exposure. Third, a key challenge in untargeted metabolomics is the difficulty of achieving Level 1 compound identification, as many detected metabolites lack reference standards for full MS/MS and retention time confirmation. Consequently, some identifications remain putative and require further validation through targeted approaches. To enhance confidence in our annotations, we incorporated an in-house reference library containing approximately 300 authentic chemical standards, which enabled Level 1 confirmation for a subset of metabolites. The full list of these standards is provided in Appendix A. Fourth, the modest sample size, with participants exclusively from Brescia, Italy, limits the generalizability of our findings, underscoring the need for larger, more diverse, and longitudinal population studies to confirm biomarker relevance and clarify the temporal dynamics between exposure and disease onset. Future studies will also acknowledge individual-level factors such as BMI, diet, and variability in manganese absorption comprehensively. This reflects a broader challenge in environmental epidemiology, where diet and bioavailability are inherently difficult to quantify precisely in human populations. Additionally, these metrics will be used to inform power calculations to ensure future studies are statistically robust and better equipped to account for individual variability. Moreover, more granular exposure assessments incorporating repeated biological measurements will improve the accuracy of exposure characterization. Lastly, to strengthen mechanistic insight, experimental validation in targeted cellular and animal models will be critical for confirming causal relationships and elucidating the biological processes underlying manganese-associated neurodegeneration. Together, these improvements will support the development of refined biomarkers and targeted strategies for prevention and intervention.

## 5. Conclusions

In conclusion, this study underscores the dangers of environmental Mn exposure, particularly in industrial regions. The annotation of metabolites like 3-sulfoxy-L-tyrosine, formiminoglutamic acid, glyoxylic acid, glycocholic acid, butanoate, glutamate, and palmitelaidic acid could pave the way for improved screening protocols and targeted interventions aimed at mitigating the neurotoxic effects of Mn. Additionally, our findings highlight distinct metabolomic and lipidomic alterations induced by Mn exposure compared to idiopathic Parkinson’s disease, particularly in pathways related to glucose homeostasis, amino acid metabolism, butanoate metabolism, and lipid signaling. These perturbations may reflect the combined neurotoxic impact of Mn on excitotoxicity, oxidative stress, and neuroinflammation, which could identify MnIP. Recognizing these metabolic differences could aid in the development of more precise biomarkers for early detection and differentiation of MnIP from idiopathic Parkinson’s disease. Finally, elucidating the metabolic pathways disrupted by Mn exposure may inform therapeutic approaches to prevent or slow PD progression in at-risk populations.

## Figures and Tables

**Figure 1 metabolites-15-00487-f001:**
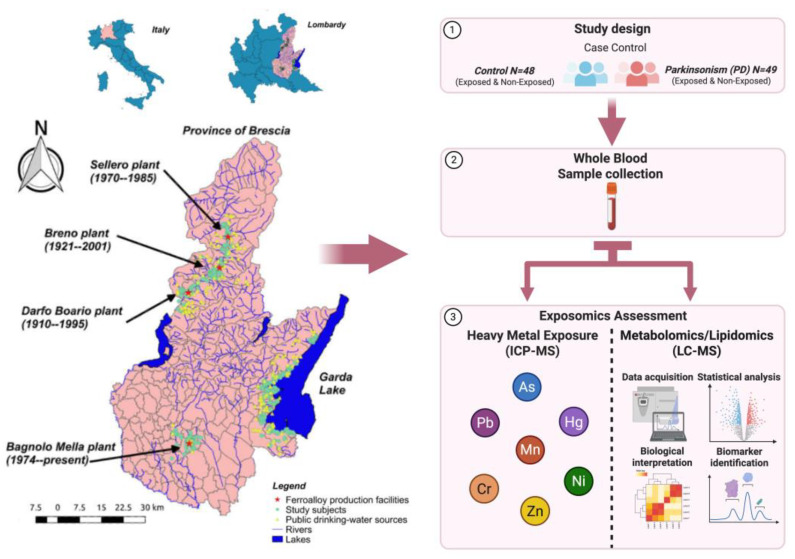
Study design and geographic overview of the Parkinsonism case–control population in the Province of Brescia, Italy.

**Figure 2 metabolites-15-00487-f002:**
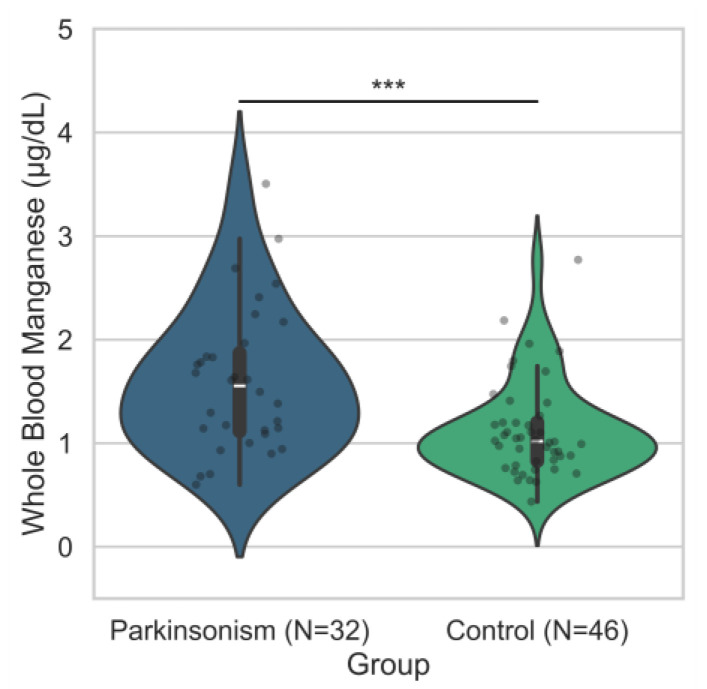
Distribution of whole-blood manganese concentrations by disease status. Violin plots show the distribution of whole-blood manganese levels (µg/dL) among individuals with Parkinsonism (*n* = 32) and controls (*n* = 46). The median manganese concentration was higher in the Parkinsonism group (1.55 µg/dL; IQR = 0.75) compared to the control group (1.02 µg/dL; IQR = 0.37). A Mann–Whitney U test indicated a statistically significant difference between groups (*p* = 0.0010). *** *p*-Value ≤ 0.001.

**Figure 3 metabolites-15-00487-f003:**
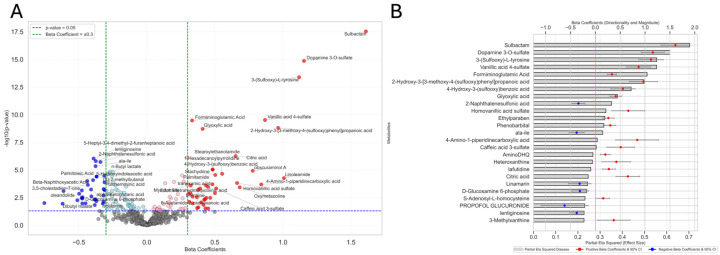
Metabolites associated with disease effect identified by ANCOVA. (**A**) Volcano plot displaying metabolite associations with disease status. The x-axis shows beta coefficients (effect size and direction), while the y-axis shows the (−log10(*p*-value)). Red points represent significantly upregulated metabolites (positive beta coefficients), and blue points represent significantly downregulated metabolites (negative beta coefficients). Light-pink and light-blue points indicate metabolites with significant *p*-values but smaller effect sizes (<|0.3|). Gray points are not statistically significant. The vertical green dashed lines mark beta coefficient thresholds of ±0.301, and the horizontal blue dashed line represents the significance threshold (*p* = 0.05). (**B**) Bar chart showing the top 25 metabolites ranked by partial eta squared (η^2^_p_), representing the proportion of variance in metabolite levels explained by disease status. Gray bars reflect this explained variance. Overlaid red and blue circles represent beta coefficients (positive and negative, respectively), with error bars indicating 95% confidence intervals (CIs). All metabolites shown passed statistical thresholds for inclusion.

**Figure 4 metabolites-15-00487-f004:**
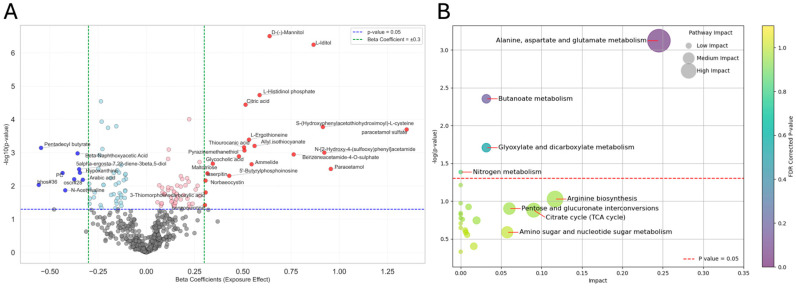
Metabolites associated with exposure effect identified by ANCOVA. (**A**) Volcano plot displaying the relationship between effect size (β coefficients) and statistical significance (−log10(*p*-value)) for individual metabolites associated with exposure effect. Each point represents a single metabolite. Red dots indicate metabolites significantly upregulated with exposure (β > 0.301, *p* < 0.05), while blue dots represent significantly downregulated metabolites (β < −0.301, *p* < 0.05). Light-pink and light-blue dots represent metabolites significant by *p*-value but with lower effect sizes. The vertical green dashed lines mark the beta coefficient thresholds of ±0.301, and the horizontal blue dashed line indicates the significance threshold of *p* = 0.05. (**B**) Pathway enrichment and topology analysis for exposure-effect-associated metabolites. Each bubble represents a metabolic pathway, where the x-axis indicates pathway impact based on relative-betweenness centrality and the y-axis shows statistical significance (−log10(*p*-value)) from a hypergeometric overrepresentation test. Bubble size reflects pathway impact, and color intensity corresponds to the FDR-corrected *p*-value. Pathways above the red dashed line (−log10(*p*) > 1.3, corresponding to *p* < 0.05) are considered significantly enriched. Analysis was performed using MetaboAnalyst 6.0 with Homo sapiens-specific KEGG pathway libraries.

**Figure 5 metabolites-15-00487-f005:**
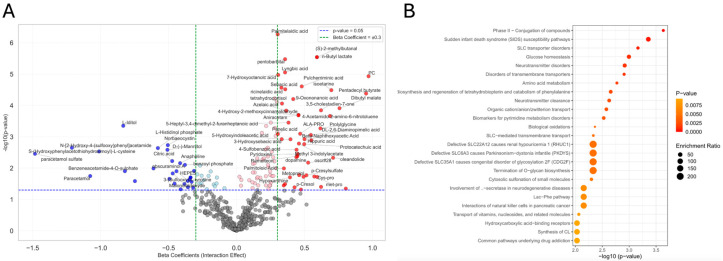
Metabolites associated with interaction effect identified by ANCOVA. (**A**) Volcano plot illustrating the interaction effect on metabolites, displaying beta coefficients (x-axis) versus (−log10(*p*-values)) (y-axis). Red dots represent metabolites significantly upregulated in the interaction group (β > 0.301, *p* < 0.05), and blue dots indicate significant downregulation (β < −0.301, *p* < 0.05). Horizontal and vertical dashed lines represent *p*-value and beta coefficient thresholds, respectively. Annotated metabolites reflect those passing both thresholds and among the most significant based on *p*-value. (**B**) Bubble plot representing the top 25 enriched metabolite sets associated with the interaction effect. Bubble size corresponds to the enrichment ratio, and color represents statistical significance (*p*-value), with darker colors indicating stronger enrichment. Pathways are ranked by (−log10(*p*-value)), with the most statistically enriched pathways appearing at the top.

**Figure 6 metabolites-15-00487-f006:**
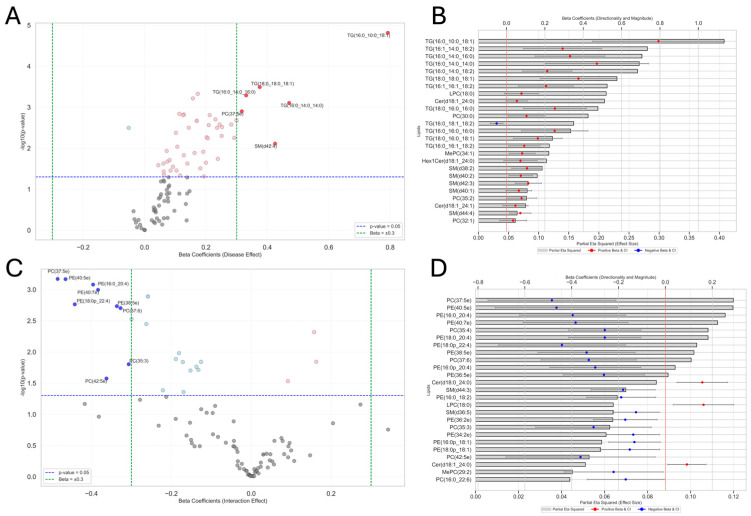
Lipids associated with disease effect and interaction effect identified by ANCOVA. (**A**) Volcano plot illustrating the disease effect on lipid species, where each point represents its associated beta coefficient (x-axis) and its (−log10(*p*-value)) (y-axis). Significant positive associations (β > 0.301, *p* < 0.05) are shown in red and significant negative associations in blue (β < −0.301, *p* < 0.05). Dashed vertical green lines indicate β = ± 0.301, and the horizontal dashed blue line indicates the significance threshold of *p* = 0.05. (**B**) Bar chart of the top 25 lipid species associated with the disease effect, ranked by partial eta squared (η^2^_p_), indicating effect size. Overlaid points reflect the direction and magnitude of beta coefficients with their 95% confidence intervals (CIs)colored by sign of association — red for positive associations, blue for negative. (**C**) Volcano plot showing lipid species associated with the interaction effect, following the same plotting conventions as Panel A. (**D**) Bar chart of the top 25 lipid species associated with the interaction effect, following the same plotting conventions as Panel B.

**Table 1 metabolites-15-00487-t001:** Demographic and lifestyle characteristics of the study population.

	*N* (%) or Mean (SD)				Parkinsonism (PD) Vs. Controls *p*-Value^a^
Characteristiss	Parkinsonism (PD)		Controls		
	Exposed	Non-Exposed	Exposed	Non-Exposed	
Sample Size (n=)	23	26	25	23	
Age (years)	70.4 (10.8)	66.4 (9.6)	70.6 (12.3)	72.4 (7.3)	*p* = 0.11
Sex					*p* = 0.15
Male	13 (56.5%)	21 (80.7%)	12 (48%)	13 (56.5%)	
Female	10 (43.4%)	5 (19.2%)	13 (52%)	10 (43.4%)	
Coffee Consumption					*p* = 1.0
Yes	21 (91.3%)	25 (96.1%)	23 (92%)	23 (100%)	
No	2 (8.6%)	1 (3.8%)	2 (8%)	0 (0%)	
Alcohol Consumption					*p* = 0.40
Yes	14 (60.8%)	19 (73%)	14 (56%)	13 (56.5%)	
No	9 (39.1%)	7 (26.9%)	11 (44%)	10 (43.4%)	
Smoking Status					*p* = 0.16
Yes	10 (43.4%)	18 (69.2%)	10 (40%)	9 (39.1%)	
No	13 (56.5%)	8 (30.7%)	15 (60%)	14 (60.8%)	
Diabetes					*p* = 1.0
Yes	3 (13.0%)	6 (23.1%)	1 (4.0%)	7 (30.4%)	
No	20 (87.0%)	20 (76.9%)	24 (96.0%)	16 (69.6%)	
Stroke					*p* = 0.49
Yes	1 (4.3%)	1 (3.8%)	0 (0.0%)	0 (0.0%)	
No	22 (95.7%)	25 (96.2%)	25 (100.0%)	23 (100.0%)	
Hypertension					*p* = 0.75
Yes	13 (56.5%)	14 (53.8%)	13 (52.0%)	16 (69.6%)	
No	10 (43.5%)	12 (46.2%)	12 (48.0%)	7 (30.4%)	
Leukemia					*p* = 0.49
Yes	0 (0.0%)	0 (0.0%)	0 (0.0%)	1 (4.3%)	
No	23 (100.0%)	26 (100.0%)	25 (100.0%)	22 (95.7%)	
Heart Disease					*p* = 0.35
Yes	8 (34.8%)	5 (19.2%)	8 (32.0%)	10 (43.5%)	
No	15 (65.2%)	21 (80.8%)	17 (68.0%)	13 (56.5%)	
Liver Disease					*p* = 0.27
Yes	1 (4.3%)	1 (3.8%)	5 (20.0%)	0 (0.0%)	
No	22 (95.7%)	25 (96.2%)	20 (80.0%)	23 (100.0%)	
Kidney Disease					*p* = 0.62
Yes	2 (8.7%)	1 (3.8%)	0 (0.0%)	1 (4.3%)	
No	21 (91.3%)	25 (96.2%)	25 (100.0%)	22 (95.7%)	
Thyroid Disease					*p* = 0.68
Yes	1 (4.3%)	1 (3.8%)	0 (0.0%)	3 (13.0%)	
No	22 (95.7%)	25 (96.2%)	25 (100.0%)	20 (87.0%)	
Geographical Site					
Bagnolo Mella	2 (8.6%)	0 (0%)	1 (4%)	0 (0%)	
Garda Lake	0 (0%)	15 (57.6%)	0 (0%)	5 (21.7%)	
Val Camonica	21 (91.3%)	0 (0%)	24 (96%)	0 (0%)	
Brescia City	0 (0%)	11 (42.3%)	0 (0%)	18 (78.2%)	

a *T*-test (age) and chi-squared
$X^{2}$ or Fisher’s exact tests as appropriate (coffee consumption, alcohol consumption, smoking status, individual comorbidities).

**Table 2 metabolites-15-00487-t002:** Summary statistics for whole-blood manganese concentrations in parkinsonism and control groups.

*Group*	N	Mean	SD	Median	IQR	Min	Max	Parkinsonism (PD) Vs. Controls *p*-Value
Parkinsonism (PD)	32	1.6	0.7	1.55	0.75	0.6	3.5	0.001
*Control*	46	1.12	0.46	1.02	0.37	0.44	2.77	

Values are expressed in µg/dL. Comparison between groups was performed using a Mann–Whitney U test. *N* = number of participants per group.

**Table 3 metabolites-15-00487-t003:** Conditional logistic regression results for whole-blood manganese concentrations and Parkinsonism status.

Variable	β Coefficient	Odds Ratio (OR)	95% Confidence Interval	*p*-Value
Whole Blood Mn (µg/dL)	0.88	2.42	1.13–5.17	0.022

Analysis based on 32 matched case–control pairs (*n* = 64), with matching on age and sex. Results reflect the association between whole-blood manganese levels (µg/dL) and Parkinsonism diagnosis. Matching variables were accounted for through the stratified design of the conditional logistic regression.

**Table 4 metabolites-15-00487-t004:** Summary of significant metabolites, lipids, and biologically relevant pathways for disease, exposure, and interaction effects.

Omic	Disease Effect	Exposure Effect	Interaction Effect
Metabolomics	3-sulfoxy-L-tyrosineosine	glycocholic acid	palmitelaidic acid
	formiminoglutamic acid	butanoate	vitamin B6 metabolism
	glyoxylic acid	glutamate	glucose homeostasis
	amino acid metabolism	alanine, aspartate, and glutamate metabolism	amino acid metabolism
	citrate cycle (TCA cycle)	butanoate metabolism	SLC transporters disorders
		SLC-mediated transmembrane transport	
Lipidomics	triacylglycerols	ceramides	phosphatidylethanolamines
	ferroptosis		ferroptosis
	endocannabinoid signaling		endocannabinoid signaling
			sphingolipid metabolism and signaling

## Data Availability

The original contributions presented in this study are included in the article/Appendix A. Further inquiries can be directed to the corresponding author.

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
