# Peer review of "Exploratory Metabolomic and Lipidomic Profiling in a Manganese-Exposed Parkinsonism-Affected Population in Northern Italy"

_metabolites, 2025, doi:10.3390/metabo15070487_

Round 1

Reviewer 1 Report

Comments and Suggestions for Authors

This study provides important exploratory insights into metabolomic and lipidomic changes in a manganese-exposed Parkinsonism population, though some queries need to be addressed while revising the manuscript.

  • What were the specific inclusion and exclusion criteria used to select participants within the exposed and non-exposed groups, beyond geographic residence and length of domicile? Were any individuals excluded based on occupational exposure, medical conditions, or other confounding factors?
  • The sample size is very small (N=97). How did the authors address potential limitations in statistical power and ensure the robustness of the metabolomic and lipidomic findings?
  • Could the authors clarify which type of anticoagulant was used during whole blood collection (e.g., EDTA, heparin, or citrate), given its potential impact on downstream metabolomic and lipidomic analyses?
  • Adding a brief overview or schematic of the experimental workflow would help readers better understand the overall study design and methodology.
  • Were the associations between manganese levels and Parkinsonism consistent across both exposed and non-exposed groups?
  • Could the authors clarify whether the observed metabolomic and lipidomic alterations were analysed separately within the exposed and non-exposed groups, or were the effects primarily driven by one subgroup?
  • It is recommended that the authors validate the key identified metabolites and lipids. Using targeted analyses or authentic standards would enhance confidence in the reliability of the reported findings.

Reviewer 2 Report

Comments and Suggestions for Authors

First of all, although the study was based on a case-control design, the sample size (N = 97) is limited for such high-dimensional omics data, which carries the risk of increasing false-positive rates, especially in statistical analyses where multiple comparisons are made. Although FDR corrections were applied, some interpretations of biological significance should be approached with caution, as some of them were based on pathways that did not show significance after FDR correction. Furthermore, since the study focused on an environmental exposure (manganese) specific to the Brescia region, the generalizability of the results is limited. Although the exposure classification was based on GIS-based location information and the condition of living in the same region throughout the lifespan, real-time Mn bioavailability and behavioral factors (e.g. workplace exposure, diet) at the individual level were not controlled. This poses the risk of ignoring the heterogeneity of exposure levels among individuals. Furthermore, measuring blood Mn levels at a single time point may not adequately reflect chronic exposure. Although the processing, statistical analyses and pathway analyses of omic data were carried out in great detail, only basic covariates such as age and gender were considered in ANCOVA models; other potential interaction variables (e.g. nutrition, genetic polymorphisms) were not included in the model. In addition, the identification levels of metabolites associated with significant findings were mostly Level 2 and below, indicating that structure validation was incomplete and may limit the translational potential of biomarkers. Finally, although the significances found for interaction effects are biologically meaningful, no forward plan was presented for testing these effects at the mechanistic level or for experimental validation. Despite all these limitations, the study provides an important and original contribution to the explanation of environmental neurotoxic exposures with a systems biology approach. In addition, the similarity rate of the article should definitely be reduced.

Comments on the Quality of English Language

minor editing required

Reviewer 3 Report

Comments and Suggestions for Authors

The manuscript by Lewis et al (Metabolites) investigates lipidomic and metabolic assessment in whole blood of healthy and parkinsons disease patients exposed to different levels of manganese.  Overall, the concept of the paper is interesting however several factors limit the scientific value of the data interpretation and findings.

Major Concerns:

  1. Although the authors claim that the study was adequately powered, this does not seem to agree with what is accepted in the literature regarding confounders in metabolomics and lipidomics research. Most importantly, the manuscript by Dunn et al Metabolomics 2015 11:9-26, most notably figure 3, highlights that you need around 200-400 individuals to achieve a 85% confidence in separating out demographic factors for metabolomics data.  Thus, if groups exist with different age, BMI, gender, etc, medication status, all that is really being characterized is demographic factors if things are imbalanced.
  2. Since the hypothesis is that manganese is correlated with metabolites or lipids, to impact Parkinsons Disease, was individual analytes correlated wit manganese levels?
  3. Where the patients medicated in healthy? Or Parkinson Disease? If they were medicated, which medication Carbodopa, amantadine, MOA-B inhibitors?  What does?  What other medication, statins, diabetes medication?  All of this will impact the metabolome and lipidome.  Further, it looks like comorbities were bundled together, it would be important to separate those, also if they were self reported or diagnosed clinically?  Further in Figure 2, a lot of the increased metabolites are pharmaceuticals or break down products of dopa, thus questioning that these analytes might be purely associated based on treatment.  Figure 3 – you have analytes such as mannitol, which are impacted by diet.  Further the fact that bile acids are mentioned require an assessment of BMI or gender association.
Comments on the Quality of English Language

Overall, the manuscript is well written but could be improved in some section, mainly the introduction simplifying the narrative.

Round 2

Reviewer 1 Report

Comments and Suggestions for Authors

The author has successfully addressed the queries and revised the manuscript accordingly. I recommend the manuscript for publication with one suggestion: the author may consider including the results of the Two-Way Analysis of Covariance (ANCOVA) used to evaluate the association between manganese levels and Parkinsonism in both exposed and non-exposed groups as supplementary data. This would enhance the transparency and robustness of the findings.

Author Response

Comment 1: The author has successfully addressed the queries and revised the manuscript accordingly. I recommend the manuscript for publication with one suggestion: the author may consider including the results of the Two-Way Analysis of Covariance (ANCOVA) used to evaluate the association between manganese levels and Parkinsonism in both exposed and non-exposed groups as supplementary data. This would enhance the transparency and robustness of the findings.

Response 1: Thank you for your valuable suggestion. The results of the Two-Way Analysis of Covariance (ANCOVA) examining the association between manganese levels and Parkinsonism in both exposed and non-exposed groups have now been included in Supplemental Table 2. The manuscript has been revised accordingly, and we appreciate your recommendation to enhance the transparency and robustness of the findings.

Reviewer 2 Report

Comments and Suggestions for Authors

I have received sufficient responses to my comments and the article is ready for publication.

Author Response

Comment 1: I have received sufficient responses to my comments, and the article is ready for publication.

Response 1: Thank you for your support of the publishing of this manuscript and your guidance throughout the review process. 

Reviewer 3 Report

Comments and Suggestions for Authors

The authors responses have further highlighted the flaws in the study challenging the scientific value of the portrayed results

Author Response

Comment 1: The authors responses have further highlighted the flaws in the study, challenging the scientific value of the portrayed results. 

Response 1: Thank you for your guidance throughout this review process.